# Beyond Care: A Scoping Review on the Work Environment of Oncology Nurses

**DOI:** 10.3390/nursrep15090324

**Published:** 2025-09-05

**Authors:** Asia Vailati, Ilaria Marcomini, Martina Di Niquilo, Andrea Poliani, Debora Rosa, Giulia Villa, Duilio Fiorenzo Manara

**Affiliations:** 1Center for Nursing Research and Innovation, Faculty of Medicine and Surgery, Vita-Salute San Raffaele University, 20132 Milan, Italy; vailati.asia@hsr.it (A.V.); m.diniquilo@studenti.unisr.it (M.D.N.); poliani.andrea@unisr.it (A.P.); rosa.debora@unisr.it (D.R.); villa.giulia@hsr.it (G.V.); manara.duilio@hsr.it (D.F.M.); 2Department of Obstetrics and Gynaecology, IRCCS Ospedale San Raffaele, 20132 Milan, Italy

**Keywords:** nursing work environment, oncology nursing, scoping review, organizational factors

## Abstract

**Background:** The Nursing Work Environment (NWE) plays a critical role in determining the quality of care, staff well-being, and organizational performance, particularly in oncology settings. Despite increasing attention, a comprehensive synthesis of organizational factors shaping oncology NWEs has been lacking. This scoping review aimed to describe the key features of oncology NWEs and to explore the outcomes associated with these characteristics. **Methods**: A scoping review was conducted following the Joanna Briggs Institute guidelines. Peer-reviewed studies published in English or Italian were included without time restrictions. Literature searches were performed in MEDLINE via PubMed, CINAHL, and Scopus between January and April 2025. **Results**: Twenty studies met the inclusion criteria. Key organizational characteristics of oncology NWEs were grouped into the following four domains: leadership and organizational support; workload and resource availability; ethical climate and collegial relationships; and physical and structural conditions of care settings. Across the studies, a positive NWE was frequently reported to be associated with improved nurse-related outcomes and, to a lesser extent, with patient-related outcomes. However, these associations should be interpreted with caution due to the heterogeneity of contexts and the predominance of cross-sectional designs. **Conclusions**: The NWE is a strategic element in delivering effective, safe, and sustainable oncology care. Practical actions for nurse managers and healthcare leaders include implementing leadership training programs, ensuring adequate staffing and resource allocation, fostering open communication, and promoting interdisciplinary collaboration. These measures are essential to protect staff well-being and guarantee high-quality, patient-centered care.

## 1. Introduction

In recent years, the concept of the Nursing Work Environment (NWE) has attracted growing attention in the literature, especially for its relevance to care quality and organizational well-being in the evolving field of health sciences. The NWE was defined as the set of organizational features within the workplace that may enhance or constrain professional nursing practice [1]. To better conceptualize the complexity of the NWE, its characteristics can be grouped into four key dimensions. The first dimension concerns the nature of the work, including role clarity and workload, as well as aspects such as professional autonomy, adequate staffing, and nurse-physician collaboration [2]. The second dimension relates to the relational and social context, encompassing teamwork, interprofessional interactions, and peer support [3,4]. The third dimension addresses the structural and physical aspects of the care setting, such as safety, space organization, and access to resources. Finally, organizational culture, defined as a shared system of values, norms, and practices, serves as a cross-cutting element that shapes the quality of the work environment [5,6].

Numerous studies have highlighted the crucial role of the NWE in shaping patient outcomes. Findings from these studies suggested that a positive NWE was associated with improved clinical results, including lower mortality rates and a reduction in adverse events such as nosocomial infections [7], patient falls, medication errors, pressure ulcers, and failure-to-rescue incidents [7,8]. Furthermore, a well-structured and supportive NWE contributes to a stronger culture of patient safety, an increased perception of care quality [9], shorter lengths of hospital stay, and a lower rate of hospital readmissions [10].

These considerations become particularly relevant in oncology settings, where nurses face a high emotional and clinical burden [11]. A defining element of the oncology work environment is the centrality of the therapeutic relationship. Frequent and prolonged contact with patients leads to the development of intense relationships that require listening, empathy, emotional availability, and continuity of care. Over time, these relationships can become significant sources of stress and concern [4]. Therefore, a work environment that encourages reflection, emotional sharing, and support is essential. Additionally, empathetic management of communication with patients and their families is crucial and requires advanced skills, as well as opportunities for peer discussion [11,12]. Effective oncology care also relies on multidisciplinary integration, which is based on effective communication, clear roles, and mutual respect [1,3]. The complexity of oncology care demands structured, cohesive, and supportive work environments [11,13,14].

Given the critical and emotionally intense nature of oncology nursing, a comprehensive understanding of the organizational characteristics of NWE is essential. While the relevance of the NWE has been extensively documented in general healthcare, evidence in oncology remains fragmented and lacks a systematic synthesis. This scoping review addresses this gap by mapping and organizing the available evidence on the oncology NWE, thereby providing insights to guide nurse managers and healthcare leaders in creating supportive and sustainable work environments.

## 2. Methods

Scoping reviews are commonly carried out to evaluate the relevance of a given topic within the literature and to provide a comprehensive overview of existing studies. They represent a popular approach for synthesizing research in areas where robust evidence is limited, to rapidly map the key concepts that define and underpin the field of inquiry [15]. Our scoping review aimed to: (a) describe the main features of the work environments of nurses working in oncology contexts and (b) explore which outcomes are associated with these environmental characteristics.

To comprehensively examine the literature on the work environment of oncology nurses, the following research questions were formulated:

What characteristics of the nursing work environment have been studied in oncology settings?What outcomes are associated with the features of oncology nursing work environments?

To conduct this study, a scoping review was carried out following the Joanna Briggs Institute (JBI) Manual for Evidence Synthesis [16]. The PRISMA-ScR checklist (Appendix A) was used to report the Scoping Review.

The research questions were developed using the Population, Concept, and Context (PCC) framework [16] (Table 1).

### 2.1. Eligibility Criteria

#### 2.1.1. Inclusion Criteria

The inclusion of studies was based on their relevance to the research questions and their adherence to the PCC framework [16]. Participants were considered eligible if they were nurses or patients, with no restrictions on age, gender, cultural background, or social circumstances.

Articles were selected for inclusion in this scoping review without time restrictions to ensure a comprehensive exploration of the literature. Additionally, no geographical limitations were applied, covering various settings such as hospitals, communities, and clinics, thus encompassing all healthcare systems. Both primary and secondary, qualitative and quantitative sources of literature were included. Primary studies already incorporated in secondary reviews were considered only when, within those reviews, they were analyzed with an objective different from our research question, thereby minimizing the risk of overlap.

#### 2.1.2. Exclusion Criteria

Papers written in languages other than English and Italian were excluded. Gray literature was also excluded, as the aim of this review was to provide an overview of the most established and reliable evidence available in the peer-reviewed scientific literature. This choice ensured methodological rigor and comparability across studies.

#### 2.1.3. Limits

A language filter was used to select records written in English or Italian.

### 2.2. Search Strategies

A three-stage search was conducted in accordance with the JBI guidelines [16]. The entire selection process followed the PRISMA Flow Diagram [17]. The literature search was performed between January and April 2025. An initial exploratory search was conducted in MEDLINE via PubMed, and CINAHL, followed by refinement and expansion of the strategy. Keywords and index terms were identified by analyzing the titles and abstracts of relevant articles. These terms informed the construction of the final search strategy. Search strategies were developed using a combination of indexed terms (e.g., MeSH and CINAHL Subject Headings) and free-text terms derived from the PCC. Boolean operators (AND, OR) were used to combine terms and ensure comprehensive coverage of the topic. Following the refinement of the search strategy, a final search was performed in the Scopus database to identify any additional relevant literature. In the final phase, the reference lists of all included studies and reports were screened to identify further relevant publications. To ensure methodological rigor, the entire search process was designed and refined in collaboration with a university librarian (Appendix A).

### 2.3. Document Selection

The selection of relevant articles was conducted though the Rayyan software platform in three phases: (a) an initial screening of titles and abstracts; (b) full-text retrieval and screening; (c) data extraction and comparison of the collected information. The entire selection process, including title and abstract screening and full-text selection, was carried out independently by two reviewers (A.V., I.M.). Any discrepancies in record inclusion were resolved through discussion and, when necessary, with the assistance of a review expert (G.V.).

### 2.4. Data Extraction

Metadata were exported from Zotero (Version 7.0.24) and manually verified upon import. In line with JBI guidelines, a comprehensive data-extraction matrix was developed encompassing: (a) general information (title, authors, year of publication); (b) study aims and objectives; (c) study design; (d) statistical methods and measurement scales; (e) workplace environment characteristics; (f) outcomes and key findings (Appendix A).

### 2.5. Results Presentation

The results were described in a narrative form and supported by tables that synthesized the evidence.

## 3. Results

A total of 990 records were found. After deduplication, 444 duplicates were removed, leaving 546 articles for title and abstract screening. Of these, 475 were excluded based on irrelevance to the inclusion criteria. A total of 71 articles were deemed eligible for full-text screening; however, five could not be retrieved even after contacting library services. The remaining 66 full-text articles were assessed in detail, resulting in the inclusion of 20 studies in the final scoping review (Figure 1).

### 3.1. Study Characteristics

The timeframe of the included studies ranged from 2004 to 2023. No eligible studies published after 2023 were identified. The studies included in the analysis were predominantly quantitative, followed by qualitative studies and literature reviews. Among the quantitative research, the most frequent were cross-sectional studies with a descriptive-correlational approach [14,18,19,20,21,22,23,24,25,26]. The qualitative studies included a focused ethnographic study [12], a thematic analysis [27], and a content analysis [13]. One study used a mixed-methods approach [28]. Additionally, several reviews were included: three integrative reviews [29,30,31], a narrative review [32], a systematic review [33], and a meta-synthesis [34].

Some studies included subgroups such as pediatric nurses [23,24]. One study also involved nursing managers, coordinators, and quality experts [13]. Participants across all studies held either a nursing diploma or a bachelor’s degree. Reported ages ranged from an average of 22 [20] to 46 [12], and work experience averaged 6.7 years, ranging from 4 to 23 years.

Most studies were conducted in hospital settings, except for one based in a university environment [23]. Units studied included oncology, hematology, intensive care, emergency departments, bone marrow transplant (adult and pediatric), palliative care, day surgery, solvents, and hospice.

### 3.2. Organizational Characteristics of Oncology Work Environments

Based on the results, the findings can be organized into four key domains: leadership and organizational support, workload and resource availability, ethical climate and collegial relationships, and the structural and physical characteristics of oncology care settings. Figure 2 summarizes how many times each workplace environment characteristic was analyzed across the studies included in the scoping review, with “workload and resources availability” representing the most frequently investigated domain (10 out of 20 studies).

#### 3.2.1. Leadership and Organizational Support

In oncology care, leadership quality plays a crucial role in shaping nurses’ experiences and outcomes. Transformational leadership, characterized by active listening, emotional support, and inclusive decision-making, has been consistently linked to higher job satisfaction, reduced burnout, and more favorable perceptions of workload and resource availability [35]. Liu et al. [20] found that nurses reported a positive perception of leadership and an overall favorable view of their work environment. Leadership effectiveness showed a significant positive correlation with professional benefits and a negative correlation with transition shock [20], while Kamimura et al. [27] found that strong leadership support in outpatient oncology settings was linked to higher job satisfaction and perceived care quality. In contrast, bureaucratic or unsupportive leadership styles, often centered on cost-efficiency at the expense of staff well-being, have been associated with decreased motivation and increased turnover intentions [12,31]. Data from the Swedish component of the RN4CAST project [19], provided strong empirical evidence that poor leadership and unsupportive work environments are closely linked to higher levels of nurse burnout and job dissatisfaction. Furthermore, their analysis showed that negative perceptions of leadership, such as low ratings for clinical support and poor listening, were strongly correlated with nurses’ intention to leave.

#### 3.2.2. Workload and Resource Availability

High workload, often driven by staffing shortages, represents a significant source of stress for oncology nurses. In pediatric settings, it was identified as the second most considerable stressor, following inadequate staffing [33]. Similarly, Campos de Carvalho et al. [23] reported that nearly half of the surveyed nurses considered their workload heavy or very heavy, attributing this to multitasking, excessive administrative responsibilities, and insufficient personnel. These conditions were closely associated with reduced quality of care and heightened psychological distress.

Multiple studies have linked burnout and turnover to factors such as excessive operational demands, the inherent clinical complexity of oncology care, and the lack of sufficient time for direct patient interaction [12,24,30]. At the global level, the persistent shortage of oncology-trained nurses further represents a major barrier to maintaining consistent, high-quality care [32].

Collectively, these findings underscore the importance of maintaining manageable workloads and ensuring adequate staffing and resources to support oncology nurses and safeguard patient outcomes. To mitigate these challenges, structured support strategies, such as group debriefings and grief-processing interventions, have been recommended [28].

#### 3.2.3. Ethical Climate and Collegial Relationships

Peer relationships represent a key resource in helping oncology nurses manage emotional stress. Mutual support, empathetic listening, and effective team communication were consistently identified as protective factors that fostered resilience and reduced the risk of burnout. In contrast, poor interpersonal dynamics were linked to emotional distress and professional isolation [34].

The quality of interprofessional relationships also played a significant role, particularly through the ethical climate. Ventovaara et al. [21] found that nurses with no intention of leaving their jobs reported a significantly more positive ethical climate compared to those considering resignation. Only 10% of nurses intending to leave felt they could “almost always” provide care aligned with their values, versus 31% of those who remained committed, underscoring the association between ethical congruence and organizational retention.

Despite efforts to foster supportive and participatory work environments, significant challenges persist. For instance, Al-Ruzzieh et al. [18] reported that interpersonal relationships were the lowest-rated dimension within the Perceived Professional Work Environment Index (PPWEI) [36], highlighting persistent concerns around collegiality and team cohesion.

Compounding these issues, workplace violence, both verbal and physical, remains a pervasive and serious concern, often perpetrated by distressed patients or family members. As Mojarad et al. [13] observed, the absence of clear organizational protocols and inadequate support systems for handling such incidents contributes to a deteriorating ethical climate and increased emotional vulnerability.

#### 3.2.4. Physical Structure and Operational Conditions in Oncology Settings

The structural and physical characteristics of the work environment significantly influence nurses’ ability to provide effective and compassionate care in oncology settings. Nurses working in well-equipped, organized environments reported greater satisfaction with the care delivered, while disorganized or poorly resourced settings were associated with perceptions of heightened clinical risk, workflow inefficiencies, and delays in care provision [27].

Logistical barriers, such as insufficient supplies, inefficient spatial layout, and unwelcoming physical environments, were found to hinder nurses’ capacity to maintain “nursing presence” and to form therapeutic relationships with patients. These structural deficiencies compromise both the timeliness and quality of care, ultimately impacting patient satisfaction and clinical outcomes [13].

In resource-limited oncology contexts, the lack of basic infrastructure, including dedicated treatment areas, modern equipment, and safe, functional workspaces, posed a major obstacle to delivering safe and continuous care. These limitations not only increased the risk of clinical errors but also disrupted care coordination and continuity [31,32].

Additional operational challenges, such as overcrowding and the absence of private areas for emotionally sensitive discussions or staff support, further exacerbated these issues. These environmental deficits contributed to increased emotional strain, communication barriers, frustration, and ultimately lower professional satisfaction among oncology nurses [24,25].

### 3.3. Outcomes

To efficiently present the outcomes of the selected studies, it is appropriate to divide them into two categories: (a) patient-related outcomes and (b) outcomes related to nurses.

#### 3.3.1. Patient-Related Outcomes

##### Quality and Safety

The quality and safety of nursing care in oncology are recognized as essential components for both patient protection and the improvement of clinical outcomes. These dimensions are influenced by multiple interrelated factors, including clear and continuous communication among team members, the consistent and attentive presence of nurses at the bedside, and the application of standardized care procedures [12,13,27].

Among these, interprofessional communication stands out as a critical element in ensuring patient safety. It plays a key role in preventing errors in the prescription and administration of oncologic therapies and is vital for managing complex treatment plans [27].

The adoption of shared, standardized clinical-care protocols further contributes to safety by reducing interpersonal conflict, promoting consistent care delivery, and increasing healthcare professionals’ sense of security [27].

Beyond local practices, broader healthcare system factors also influence care quality. Adverse event prevention programs and the use of quality indicators are essential tools for managing patient safety risks. Institutions that implement incident reporting systems tend to show improvements in nursing care quality and reductions in clinical risk [32].

##### Missed Nursing Care

In complex oncology care settings, the phenomenon of Missed Nursing Care (MNC) has gained increasing attention. MNC refers to necessary nursing interventions that are either omitted, delayed, or performed incompletely due to organizational or systemic inefficiencies. This directly compromises the effectiveness and safety of care delivery [13,27].

A primary contributing factor is the shortage of qualified nursing staff, particularly critical in oncology, where care demands are high and increasingly complex. Staffing shortages are often the result of cost-containment strategies that replace registered nurses with underqualified personnel, resulting in more frequent errors and omissions. The most affected activities include clinical monitoring, therapeutic education for patients and caregivers, and emotional support, all of which negatively impact both care quality and patient safety [12].

Inadequate workforce management also plays a significant role. Increased workloads, extended or overtime shifts, and high turnover reduce care continuity and jeopardize the nurse–patient relationship, an essential element in oncology care [31]. Team instability correlates with more frequent omissions and less personalized care [13].

Excessive bureaucratic burden is another key issue. Documentation requirements, while necessary for legal and administrative purposes, divert time from direct patient care, contributing to professional frustration and lower perceived care quality [13].

Significantly, the perceived ethical climate strongly influences MNC rates. According to Vryonides et al. [22], environments characterized by individualism or rigid normative structures are associated with significantly higher rates of missed care.

##### Continuity of Care and the Nurse–Patient Relationship

Continuity of care and the nurse–patient relationship are foundational elements of oncology nursing. In the context of long and emotionally demanding treatment pathways, consistent, coordinated, and human-centered care is essential for ensuring effectiveness and promoting patient well-being. Continuity goes beyond the linear progression of care; it encompasses smooth transitions across care settings and requires timely information exchange, interdisciplinary collaboration, and nurse presence at critical moments [13,32].

Nursing presence, as the intentional and empathetic act of “being there” for the patient, plays a central role in this process. It includes not only physical proximity but also active listening, emotional availability, and responsiveness to complex needs. This form of presence requires clinical competence and emotional maturity, supported by an organizational context that allows time and space for relational care [13]. Through empathy, active listening, and attentiveness to patients’ individual and existential needs, nurses can deliver personalized, integrated care that goes beyond clinical intervention alone [12]. However, factors such as team instability, staffing shortages, and increasing care complexity further undermine continuity, weakening the therapeutic alliance and reducing patient satisfaction [31]. Frequent turnover and the absence of a designated reference nurse disrupt the emotional consistency that many patients rely on during cancer treatment. In contrast, care models like Primary Nursing, which assign one nurse to follow the patient throughout the care trajectory, have been shown to strengthen professional accountability and deepen the therapeutic relationship [12].

In this context, therapeutic communication becomes essential.

##### Perceived Ethical Climate and Its Impact on Care

In oncology, where ethically sensitive situations, including pain management, end-of-life decisions, and access to experimental treatments, are frequent, the presence of a strong ethical climate is essential. It significantly influences nurses’ ability to act in alignment with their professional values, thereby impacting both patient care and staff well-being [21].

The ethical climate refers to the shared perception among healthcare professionals of what is morally acceptable within the workplace. It encompasses organizational norms, values, and informal practices that guide ethical decision-making and shape interprofessional dynamics [22]. A positive ethical climate promotes accountability, fairness, and respect for human dignity, and is associated with reduced emotional distress, greater professional commitment, and lower turnover intention [21].

Importantly, the impact of ethical climate extends beyond individual perception. Supportive ethical environments encourage transparency, trust, and continuity of care, whereas climates driven by rule adherence or individualism may foster defensive practices and normalize ethically questionable behaviors, undermining both care quality and professional integrity [34].

In fragile ethical environments, moral distress tends to increase, compromising patient safety and deteriorating the therapeutic atmosphere [22]. Nurse managers play a critical role in cultivating an ethical climate through ethical leadership, open communication, and sustained team support, helping to create respectful, collaborative, and resilient care settings even under the pressure of complex oncological care [22].

Figure 3 summarizes the number of occurrences in which patients’ outcomes were examined across the studies included in the review, with missed nursing care emerging as the most frequently investigated outcome, reported in 6 out of 20 studies.

#### 3.3.2. Outcomes Related to Nurses

##### Burnout and Compassion Fatigue

Burnout and compassion fatigue are well-documented psychological outcomes in oncology settings, driven by prolonged emotional exposure, patient suffering, and organizational stressors. Burnout manifests through emotional exhaustion, depersonalization, and reduced personal accomplishment [24]. High levels of burnout were noted, especially among older professionals, and those with more experience [24].

Job satisfaction was inversely correlated with emotional exhaustion and strongly associated with turnover intention [14]. Coping strategies such as spirituality and collegial support showed protective effects [24].

Compassion fatigue arises from constant exposure to trauma, lack of emotional training, and weak psychological support programs [29]. Evidence shows a strong association between burnout and compassion fatigue, while lower professional satisfaction is linked to higher fatigue. The highest levels are typically found among less experienced nurses or those working in environments with poor organizational support. In addition, personal factors such as sleep deprivation and heavy family responsibilities further decrease compassion satisfaction [12,13].

Qualitative studies [25] confirmed that blurred boundaries between professional and personal life intensify emotional strain. Structured resilience programs, such as Circle of Care Retreats, which focus on resilience-building, grief processing, and the creation of peer support networks, have proven to be effective primary prevention strategies in emotionally demanding care settings [28].

##### Job Satisfaction

Job satisfaction in oncology nursing is shaped by multiple interrelated factors, including salary, managerial support, workload, and opportunities for professional development [33]. Moderate levels of dissatisfaction have been reported, particularly in non-Magnet hospitals and in settings characterized by insufficient resources. In a sample of 305 oncology nurses, Gi et al. [33] found that those working in non-Magnet hospitals reported significantly lower job satisfaction compared to colleagues in Magnet institutions, with more than double the risk of dissatisfaction. Importantly, adequate staffing and resources significantly reduced this risk.

Other findings confirm that dissatisfaction is often linked to poor staffing, lack of time, and high turnover intentions, with only 32% of nurses certain they would remain in oncology [18]. Across the literature, the role of leadership and positive team dynamics emerges as central in sustaining motivation and mitigating burnout among oncology nurses [27,32]. Interventions that enhance autonomy, staffing adequacy, and leadership support significantly improve satisfaction [18].

##### Intention to Leave

Intent to leave is a predictor of turnover and reflects professional dissatisfaction. Lagerlund et al. [19] found that 34.6% of Swedish oncology nurses considered leaving their job, with burnout and inadequate oncology training being key predictors. Intention to leave was significantly higher among nurses with less than two years of experience and those perceiving inadequate oncology training.

Gi et al. [33] noted a moderate to high intent to leave, with dissatisfaction and a lack of staffing as major predictors. Across studies, only 32% of nurses reported no intention of leaving oncology, while 8.2% expressed a strong intention to leave within a year, 39.5% were uncertain, and 6.3% intended to leave their current position, particularly those reporting inadequate staffing and low autonomy. Intention to leave was also more prevalent among nurses with advanced academic qualifications, suggesting a misalignment between skills and available career opportunities. Ethical climates and high moral distress levels were also associated with increased turnover intentions [21].

##### Psychosocial Well-Being and Support Strategies

Psychosocial well-being encompasses emotional balance, resilience, and professional fulfillment, all of which are essential for sustaining nurses in the demanding context of oncology care. Liu et al. [20] found that 53% of new graduate nurses reported moderate resilience, while only 14% showed high resilience. Transition shock was frequent during the shift from education to clinical practice, with resilience acting as a protective factor. Resilience was negatively correlated with transition shock and positively with perceived professional benefits. Creating supportive environments characterized by team trust, open communication, and peer support is therefore vital to reducing the psychological impact of workplace stress [34].

As part of their coping strategies, oncology nurses frequently turn to spirituality and peer relationships, which serve as important sources of strength and connection [24]. Institutional programs such as Creating a Resilient Work Environment have shown promise by offering structured opportunities for peer support, grief processing, and emotional education, ultimately enhancing both individual and collective resilience [28].

Despite these protective mechanisms, moral distress remains a persistent threat, particularly in understaffed settings with heavy workloads that hinder ethical practice. Ventovaara et al. (2023), using the Moral Distress Scale–Revised (MDS-R), reported a median score of 85 (IQR 62–115), significantly higher in women, and strongly correlated with negative perceptions of the ethical climate [21].

About nurse-related outcomes, the most frequently examined aspects concerned burnout and compassion fatigue, which were reported in 10 of the 20 included studies (Figure 4).

Figure 5 provides a graphical representation of the results.

## 4. Discussion

The current landscape of oncology is increasingly characterized by clinical and organizational complexity, driven by demographic, epidemiological, and systemic changes [33]. With cancer incidence projected to exceed 27 million new cases per year by 2040, the growing demand for care will significantly intensify pressure on healthcare systems, particularly on nursing staff [33]. This context explains the increasing academic interest devoted to the oncology work environment in recent years [25,32].

Oncology care is characterized by significant emotional and organizational demands, especially in end-of-life settings. These challenges emphasize the importance of work environments that ensure adequate resources and structures, while also fostering psychological support for staff [27]. Nurses often operate under emotionally taxing conditions and resource limitations, which exacerbate systemic issues such as staffing shortages, emotional overload, and burnout [12]. In this sense, the work environment emerges not merely as a backdrop for care delivery, but as a strategic component directly impacting care quality, patient safety, staff well-being, and retention [1,5].

To better understand how these challenges are addressed and which organizational factors are most influential, this scoping review systematically examined the existing literature on oncology nurses’ work environments.

The findings are consistent with current research needs in the oncology field, as they provide an integrated and cross-cutting perspective on a topic that, although widely studied, has largely been explored in a segmented manner. Indeed, while some studies addressed specific elements of the work environment, many contributions remained focused on isolated aspects, such as psychological well-being [25], turnover [32], or job satisfaction [34], without adopting a comprehensive or system-level perspective. Moreover, few studies employed interventional designs, and those that did often lacked methodological rigor or long-term follow-up [28].

What stands out prominently is the considerable diversity in both the geographical contexts and the characteristics of the work environments analyzed in the studies. The included research covered diverse care settings, from oncology and hematology to intensive care and outpatient clinics [13,14,18], across different geographical regions. Sample sizes ranged from large-scale surveys [19] to small-scale investigations [27]. While this diversity provides a broad and enriching overview, it also reflects the heterogeneity of healthcare systems, organizational cultures, and available resources across contexts. A further limitation is the wide variety of outcome measures employed, which complicates comparability across studies and hinders the accumulation of robust evidence. Beyond this, the review also revealed that almost all studies were conducted in hospital settings, with very limited evidence from community or home oncology care. This underrepresentation highlights the urgent need to extend future research to non-hospital contexts.

The scoping review identified several organizational characteristics that shape the work environment. Our results emphasized that leadership is a central element in creating positive work environments in oncology. Transformational leadership, defined by shared vision, visible support, and presence, was associated with reduced burnout and improved job satisfaction [19,20,27]. The results confirm that transformational leadership is a central factor not only for job satisfaction but also for the prevention of burnout. This leadership style operates through multiple mechanisms: on the one hand, it fosters empowerment and trust by giving nurses a voice in decision-making processes and reducing feelings of isolation [20]; on the other hand, it contributes to a more equitable distribution of workload by valuing team members’ skills and promoting collaboration [27]. In this way, nurses experience greater control over their professional environment, a factor known to mitigate stress and turnover intentions [19]. Despite its relevance, leadership remains underexamined in its specific dimensions, particularly regarding how it should adapt to the complexity of the work environment. Moreover, while the Primary Nursing model is well established in this setting [12,35], models such as the Magnet Recognition Program [33] and the Transformational Leadership Framework [35] have rarely been tested.

A supportive work environment has a direct impact on patient outcomes, particularly regarding perceived quality, safety, and continuity of care [13,27]. Studies have identified effective communication, the continuous presence of nurses at the patient’s bedside, and the adoption of standardized procedures as key factors in preventing errors and reducing clinical risk. Notably, the phenomenon of missed nursing care is a negative indicator of the work environment, correlating with staff shortages and an unstable ethical climate [22]. Beyond organizational aspects, a significant link emerges between MNC, ethical climate, and patient safety. Studies show that work environments characterized by shared values, collaboration, and ethical support are associated with reduced care omissions, whereas settings perceived as individualistic or rigidly normative are linked to higher levels of missed nursing care [21,22]. This relationship can be interpreted through Donabedian’s framework [37], which conceptualizes quality of care as the interaction between structure (resources, organization), processes (nursing activities, ethical interactions), and outcomes (safety, satisfaction, well-being). In environments where the structure is fragile (e.g., staffing shortages, inadequate facilities) and the process is compromised (e.g., lack of ethical support and interprofessional communication), the outcomes manifest as fragmented care, increased clinical risk, and moral distress.

The work environment also directly affects oncology nurses themselves. Globally, the professional landscape is concerning, with negative perceptions of psychosocial well-being and professional stability due to increasing emotional demands and workload [24,33]. Staff retention poses a major challenge, hindered by factors such as burnout, lack of resources, limited organizational support, and low involvement in decision-making processes [14,19]. Conversely, supportive environments that value the nursing role are associated with higher retention rates and greater job satisfaction [18].

Finally, the review highlights the need to implement organizational strategies such as resilience programs, peer support groups, and spaces for emotional processing [28,29] to enhance resilience, motivation, and staff stability. These interventions are still relatively uncommon but could be considered essential to protect both the physical and mental health of the nursing workforce.

### 4.1. Implications for Practice and Research

Overall, the results of this scoping review underscore the urgent need for structural and cultural changes within oncology settings. From a practical standpoint, investing in supportive work environments is not merely a matter of staff satisfaction but a strategic imperative for improving care outcomes [38]. Organizational policies should prioritize adequate staffing levels, fair workload distribution, and meaningful inclusion of nurses in institutional decision-making processes [2]. At the same time, the near absence of evidence on community and home-based oncology nursing highlights a significant gap in the literature. Considering that a growing proportion of cancer care is now delivered in outpatient and home settings [39], future studies should explore how supportive organizational models and leadership strategies can be effectively translated beyond the hospital context.

Several management practices and organizational models have already been tested in oncology contexts and offer actionable directions for practice. The Magnet Recognition Program has been associated with higher job satisfaction and reduced turnover among oncology nurses [33], while the Primary Nursing model demonstrated benefits in terms of care continuity, professional accountability, and stronger therapeutic nurse–patient relationships [20,27]. Interventions such as the Creating a Resilient Work Environment Program, which integrate structured peer-support groups and grief-processing sessions, have shown promise in enhancing resilience and reducing emotional distress [28]. Similarly, transformational leadership frameworks, emphasizing shared vision, participatory governance, and visible managerial support, have consistently been linked with lower burnout and greater professional commitment [19,20]. These tested approaches provide concrete models that oncology institutions could adapt and implement to strengthen organizational environments.

Future research should adopt integrated and methodologically robust approaches, prioritizing interventional designs that translate evidence into practical improvements. In particular, further investigation is needed into oncology settings embedded within Magnet-recognized hospitals or those implementing Primary Nursing models, to evaluate their specific impact on work environments. Finally, a deeper examination of transformational leadership within individual oncology contexts is warranted to better understand how this approach can be effectively operationalized in diverse clinical environments. Equally important are the implications for nursing education and professional development. The complexity and emotional intensity of oncology care call for specific training in resilience, communication, interdisciplinary collaboration, and leadership. Integrating these elements into undergraduate and continuing education programs can better prepare nurses to navigate challenging environments while promoting their psychological well-being and clinical effectiveness. In particular, training on transformational leadership principles, shared governance models, and participatory approaches such as co-design and co-creation can empower nurses to contribute actively to improving their work settings [40]. To maximize impact, such programs should be aligned with international competency frameworks, such as the International Council of Nurses (ICN) Core Competencies [41] and the American Association of Colleges of Nursing (AACN) Essentials, which strengthen credibility and ensure global applicability [42].

Given the current underrepresentation of community and home oncology nursing in the literature, future practice and research should also consider these settings as priority areas.

### 4.2. Limitations

The key issue was the heterogeneity of study contexts, which ranged across various clinical settings and geographical regions, making cross-comparison difficult. Sample sizes varied widely, and most studies relied on convenience sampling, reducing the generalizability of findings. Furthermore, the review only included studies published in English and Italian, introducing a potential language bias. Gray literature was not consulted, which may have led to the omission of potentially relevant but unpublished or non–peer-reviewed studies. From a methodological perspective, there is a clear predominance of cross-sectional studies, which mainly explore associations between variables, while quasi-experimental and longitudinal designs are largely underrepresented. Future research should therefore prioritize interventional and longitudinal designs to generate more robust evidence.

## 5. Conclusions

This scoping review systematically collected and mapped the main characteristics of the work environment for nurses in oncology settings. The findings indicate that key factors are crucial in shaping both the psychosocial well-being of nurses and the quality and safety of patient care. However, the literature also highlights significant challenges, including staffing shortages, heavy workloads, insufficient organizational support, and limited nurse participation in decision-making.

Immediate priorities for practice include ensuring adequate nurse-to-patient ratios, introducing structured workload management policies, and strengthening nurses’ involvement in institutional decisions through participatory models that have already been tested in healthcare. Equally important is the integration of targeted educational initiatives into both undergraduate curricula and continuing professional development. Training programs that foster transformational leadership, effective communication, and interdisciplinary collaboration can empower nurses to navigate the complexity of oncology care.

Future research should focus on longitudinal and interventional studies to assess the effectiveness and transferability of established organizational models across oncology contexts. At the same time, it is essential to design and rigorously evaluate educational programs aimed explicitly at sustaining nurses’ resilience and professional well-being, ensuring their applicability in diverse oncological settings.

## Figures and Tables

**Figure 1 nursrep-15-00324-f001:**
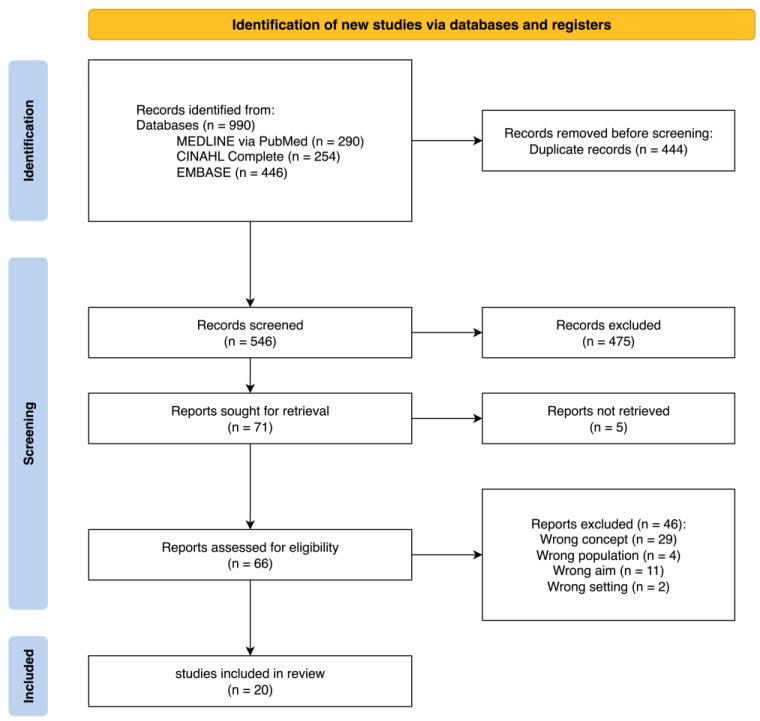
PRISMA Flow Diagram 2020 [17]: selection process.

**Figure 2 nursrep-15-00324-f002:**
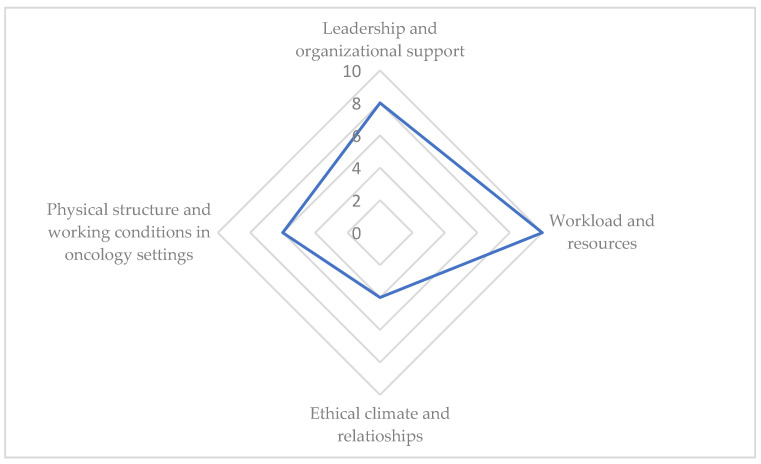
The frequency with which each work environment domain was addressed in the studies included.

**Figure 3 nursrep-15-00324-f003:**
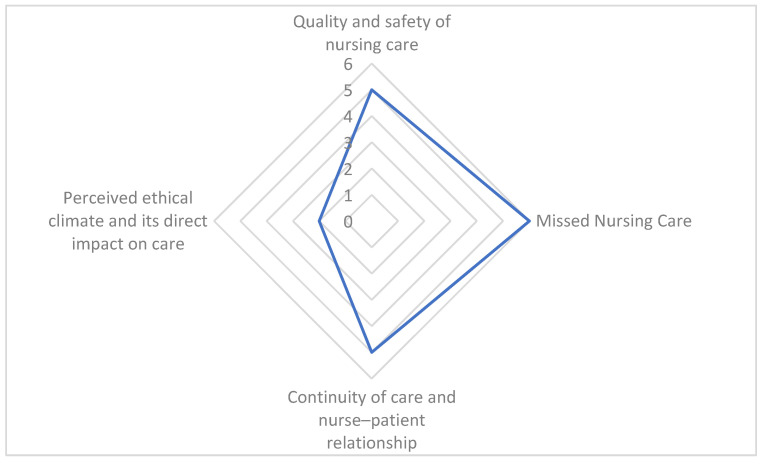
Frequency of patient outcomes examined across the included studies.

**Figure 4 nursrep-15-00324-f004:**
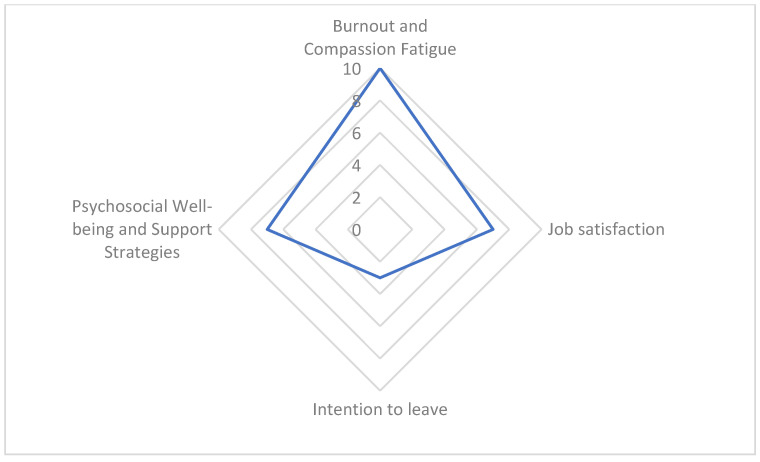
Frequency of nurse-related outcomes across the included studies.

**Figure 5 nursrep-15-00324-f005:**
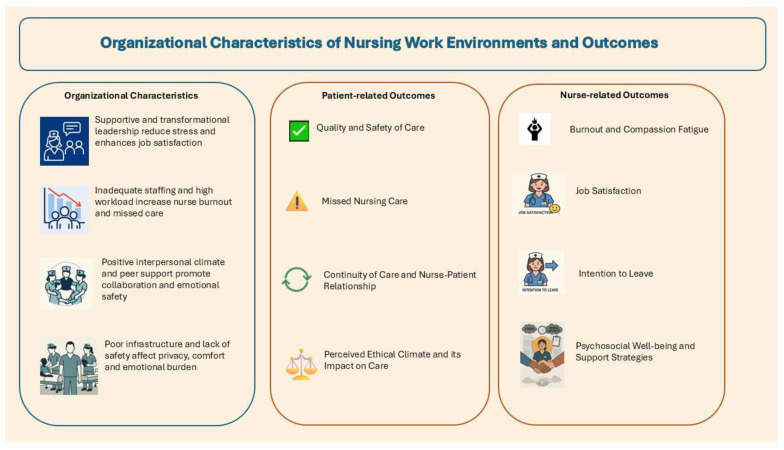
Organizational Characteristics of nursing work environment and outcomes.

**Table 1 nursrep-15-00324-t001:** PCC Framework.

PCC	
Population	Nurses working in oncology settings
Concept	Work Environments (characteristics and associated outcomes)
Context	Hospital, homes and community including cultural factors, geographic locations or gender-based interests

## Data Availability

The data presented in this study are available on request from the corresponding author.

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
