# Peer review of "Beyond Care: A Scoping Review on the Work Environment of Oncology Nurses"

_nursrep, 2025, doi:10.3390/nursrep15090324_

Round 1
Reviewer 1 Report
Comments and Suggestions for Authors
Thank you for your submission. This article addresses a highly relevant and timely topic concerning the organizational characteristics of oncology nursing work environments and their impact on both patient and nurse outcomes. The scoping review is well-conceived and grounded in current literature. However, to meet publication standards, a revision is recommended. Below are detailed suggestions to enhance the scientific rigor, clarity, and structure of the manuscript.
Considerations:
This work is very original, with as specific focus on oncology work environments, valuable analysis and underexplored issues. The contribution to healthcare improvement is high because it highlights critical organizational factors affecting care quality and workforce sustainability. The Methodological appropriateness is adequate but needs clearer articulation. The Clarity of presentation is generally good, but some sections require restructuring and refinement.
I’ll proceed with the analysis considering the several chapters of this work:
Abstract
1) The abstract summarizes the key findings and implications but lacks detail on the methodology (e.g., databases, inclusion criteria). Please clarify.
Introduction / Background
2) The background provides a solid rationale for studying oncology nursing environments by highlighting the emotional and clinical complexity of oncology care.
Recommendation: The introduction could benefit from more recent references. Please clarify the Nursing Work qEnvironment (NWE) concept (line 43 “… Nursing Work Environment (NWE)defined …” it is missing a coma).
Methodology
3) Critical Weakness:
The methodology section is present but lacks detail, such as specifics of the search strategy (keywords, Boolean operators), and Inclusion/exclusion criteria.
Recommendations: Please expand the methodology section to include all elements of the JBI scoping review protocol.
- Note: The information from Table S1 (PRISMA Checklist), Table S2 (Search Strategy), and Table S3 (Data Extraction Table) is not available.
- Line 111-112 – “An initial exploratory search was conducted in MEDLINE via PubMed, and CINAHL was conducted,…” the term “was conducted is repeated.
Results
3) Critical Weakness:
The subchapters lack a cohesive narrative or logical flow. In chapter 3.1 there are mentioned three main categories (lines 161-179) but then the authors analyzed two (chapter 3.2 - Work environment and organizational conditions (Lines 180-271); and chapter 3.3 – Outcomes (Lines 271-424).
Line 161-162: in the Chapter 3.1 is mentioned that “the objectives of the studies and grouped into three main categories: Work environment and organizational conditions in oncology settings..., … Professional development…, and… Psychological well-being and stress management”. After this, I supposed that the authors analysed each of these main categories. But then I realized this was not correct because “leadership” issues are analyzed in the “Work environment and organizational conditions…” and not in the “Professional development” as previously mentioned (line 171). This can cause some confusion and misunderstanding for the reader.
4) The figures are very useful, but they need some additional information: What is the meaning of the numbers? What type of frequency? Please add additional information.
Recommendations: Please review the Results chapter to ensure readiness and a clear understanding for the reader.
References
5) References are relevant, including some recent and foundational studies. Some corrections are needed: 1) Format inconsistencies (journal abbreviations, author lists); 2) Some entries are truncated or contain typographical errors (example: Ref 11)
Author Response
REV 1
This work is very original, with as specific focus on oncology work environments, valuable analysis and underexplored issues. The contribution to healthcare improvement is high because it highlights critical organizational factors affecting care quality and workforce sustainability. The Methodological appropriateness is adequate but needs clearer articulation. The Clarity of presentation is generally good, but some sections require restructuring and refinement.
I’ll proceed with the analysis considering the several chapters of this work:
Abstract
- The abstract summarizes the key findings and implications but lacks detail on the methodology (e.g., databases, inclusion criteria). Please clarify.
Thank you for your valuable comment. We have revised the abstract to include more detail on the methodology, specifying the databases consulted as well as the inclusion criteria applied.
Introduction / Background
2) The background provides a solid rationale for studying oncology nursing environments by highlighting the emotional and clinical complexity of oncology care.
Recommendation: The introduction could benefit from more recent references. Please clarify the Nursing Work qEnvironment (NWE) concept (line 43 “… Nursing Work Environment (NWE)defined …” it is missing a coma).
Thank you for your helpful feedback. We have updated the introduction by adding more recent references, clarified the Nursing Work Environment (NWE) concept, and corrected the punctuation as suggested.
Methodology
3) Critical Weakness:
The methodology section is present but lacks detail, such as specifics of the search strategy (keywords, Boolean operators), and Inclusion/exclusion criteria.
Recommendations: Please expand the methodology section to include all elements of the JBI scoping review protocol.
- Note: The information from Table S1 (PRISMA Checklist), Table S2 (Search Strategy), and Table S3 (Data Extraction Table) is not available.
- Line 111-112 – “An initial exploratory search was conducted in MEDLINE via PubMed, and CINAHL was conducted,…” the term “was conducted is repeated.
Thank you for your constructive comment. We have revised the methodology section to include all elements recommended in the JBI manual. The repetition of the term “was conducted” has been corrected. Keywords, detailed search strategies, and the data extraction table are now provided in the supplementary files attached to the publication (Tables S1–S3).
Results
3) Critical Weakness:
The subchapters lack a cohesive narrative or logical flow. In chapter 3.1 there are mentioned three main categories (lines 161-179) but then the authors analyzed two (chapter 3.2 - Work environment and organizational conditions (Lines 180-271); and chapter 3.3 – Outcomes (Lines 271-424). Line 161-162: in the Chapter 3.1 is mentioned that “the objectives of the studies and grouped into three main categories: Work environment and organizational conditions in oncology settings..., … Professional development…, and… Psychological well-being and stress management”. After this, I supposed that the authors analysed each of these main categories. But then I realized this was not correct because “leadership” issues are analyzed in the “Work environment and organizational conditions…” and not in the “Professional development” as previously mentioned (line 171). This can cause some confusion and misunderstanding for the reader.
You are right, and we appreciate this observation. To avoid confusion, we decided to remove the classification of study objectives that was originally introduced in Chapter 3.1, ensuring a more cohesive and logical flow. This choice is further justified by the fact that our analysis was designed to address the predefined research questions, independently of the primary objectives reported in the individual studies.
4) The figures are very useful, but they need some additional information: What is the meaning of the numbers? What type of frequency? Please add additional information.
Thank you for this helpful comment. We clarified that the numbers indicate the frequency, meaning the number of times the included studies in the scoping review analyzed each specific workplace environment characteristic.
Recommendations: Please review the Results chapter to ensure readiness and a clear understanding for the reader.
We sincerely thank you for your valuable feedback. The Results chapter has been fully revised to improve clarity, consistency, and readability.
References
5) References are relevant, including some recent and foundational studies. Some corrections are needed: 1) Format inconsistencies (journal abbreviations, author lists); 2) Some entries are truncated or contain typographical errors (example: Ref 11)
Thank you for this observation. We have carefully revised the reference list.

Reviewer 2 Report
Comments and Suggestions for Authors
Abstract
It lacks a clear and explicit statement of the main objective before the description of the methods.
In the results section, although the four identified domains are mentioned, the evidence could be better quantified (e.g., “X studies reported…”) to strengthen the synthesis.
The conclusion should be more specific regarding practical implications — for example, suggesting concrete actions that managers could implement.
Introduction
The introduction mixes concepts, impacts, and oncology-specific examples in a rather non-linear way; there is movement back and forth between general definitions and oncology-specific characteristics. It is suggested that it be organised in a logical sequence:
- Definition and importance of the Nursing Work Environment (NWE) in general.
- Evidence of the NWE’s impact on the quality and safety of care.
- Specificities and challenges of the NWE in oncology (high emotional burden, clinical complexity).
- Knowledge gap (lack of a comprehensive review on organisational factors in the oncology NWE).
Replace vague terms (“is important”, “is relevant”) with more specific statements (“directly impacts nurse retention rates and patient clinical outcomes”).
The use of references is appropriate, but some strong claims about the impact of the NWE could be supported by more recent studies (2023–2024) to reinforce currency.
The justification and relevance of the study should be further developed. What does this study add that is new?
Methods
Clarify more clearly why grey literature was excluded and discuss the possible impact of this decision.
The selection process is described, but the main text could briefly state the number of reviewers involved and how reliability was ensured (e.g., Cohen’s kappa, if calculated).
Results
The division into domains is clear and coherent, but there is repetition of ideas already presented in the introduction (e.g., relationship between leadership and burnout).
Although averages and percentages are mentioned (e.g., age, experience), many qualitative findings are not supported by aggregated quantitative data. In addition, the frequency of characteristics is shown in figures but without absolute numbers in the text. Proportions or counts should be presented in the text (“The most studied domain was leadership and organisational support, present in 10 of the 20 studies”). Also include ranges and variations where available (e.g., range of years of experience, burnout levels by setting).
Figures have axes and captions that are not very explanatory; they force the reader to return to the text to understand their meaning. Self-explanatory titles are recommended (e.g., “Figure 2 – Frequency with which each work environment domain was addressed in the 20 studies analysed”). Add more complete captions, explaining what each category means.
Discussion
There is good articulation between results and existing literature, but discrepancies between studies and possible reasons for contextual differences could be explored further.
There is a lack of discussion on the methodological quality of the included studies (even in a scoping review, it is possible to comment on robustness and limitations).
Implications for practice could be illustrated with concrete management practices and models already tested.
Conclusion
The conclusion summarises the findings well, but could indicate immediate priorities for implementation and key topics for future research. Suggestions such as “adopting participatory models” or “reorganising workload” are broad; it would be more useful to indicate specific tools, frameworks, or policies.
Author Response
REV 2
Abstract
It lacks a clear and explicit statement of the main objective before the description of the methods. In the results section, although the four identified domains are mentioned, the evidence could be better quantified (e.g., “X studies reported…”) to strengthen the synthesis. The conclusion should be more specific regarding practical implications — for example, suggesting concrete actions that managers could implement.
Thank you. We have revised the abstract: the main objective is now explicitly stated; the Conclusion has been strengthened with concrete practical implications and suggested managerial actions. We did not provide the exact number of studies per group, as the overlap between domains would make such quantification potentially misleading and create confusion in the interpretation of findings.
Introduction
The introduction mixes concepts, impacts, and oncology-specific examples in a rather non-linear way; there is movement back and forth between general definitions and oncology-specific characteristics. It is suggested that it be organised in a logical sequence:
- Definition and importance of the Nursing Work Environment (NWE) in general.
- Evidence of the NWE’s impact on the quality and safety of care.
- Specificities and challenges of the NWE in oncology (high emotional burden, clinical complexity).
- Knowledge gap (lack of a comprehensive review on organisational factors in the oncology NWE).
Thank you for your valuable comment. The introduction has been reorganized according to the reviewer’s suggestions.
Replace vague terms (“is important”, “is relevant”) with more specific statements (“directly impacts nurse retention rates and patient clinical outcomes”).
Thank you. We replaced vague terms in the introduction section.
The use of references is appropriate, but some strong claims about the impact of the NWE could be supported by more recent studies (2023–2024) to reinforce currency.
Thank you for your helpful feedback. We have updated the introduction by adding more recent references
The justification and relevance of the study should be further developed. What does this study add that is new?
We thank the reviewer for the insightful comment. The rationale of the study has been revised and further developed to better emphasize its justification and relevance.
Methods
Clarify more clearly why grey literature was excluded and discuss the possible impact of this decision.
We thank the reviewer for the useful suggestion. The text has been revised to clarify the rationale for excluding grey literature and to acknowledge the potential impact of this decision, as recommended.
The selection process is described, but the main text could briefly state the number of reviewers involved and how reliability was ensured (e.g., Cohen’s kappa, if calculated).
We thank the reviewer for this useful comment. The section has been revised to specify that the selection process was conducted independently by two reviewers, with discrepancies resolved through discussion and, when necessary, with the involvement of a review expert. Although Cohen’s kappa was not calculated, the use of independent screening and consensus ensured methodological rigor and reliability.
Results
The division into domains is clear and coherent, but there is repetition of ideas already presented in the introduction (e.g., relationship between leadership and burnout).
Although averages and percentages are mentioned (e.g., age, experience), many qualitative findings are not supported by aggregated quantitative data. In addition, the frequency of characteristics is shown in figures but without absolute numbers in the text. Proportions or counts should be presented in the text (“The most studied domain was leadership and organisational support, present in 10 of the 20 studies”). Also include ranges and variations where available (e.g., range of years of experience, burnout levels by setting).
We sincerely thank you for this constructive feedback. In response, the Results section has been carefully revised. Quantitative data have been more consistently integrated to support qualitative findings. Absolute numbers have been added in the text to complement figures and ensure greater clarity and completeness.
Figures have axes and captions that are not very explanatory; they force the reader to return to the text to understand their meaning. Self-explanatory titles are recommended (e.g., “Figure 2 – Frequency with which each work environment domain was addressed in the 20 studies analysed”). Add more complete captions, explaining what each category means.
We thank the reviewer for the suggestion. The figures have been revised with self-explanatory titles and more detailed captions. To improve clarity and readability, the figures related to outcomes have been moved to the end of the Results section, so that they can be interpreted more easily in relation to the corresponding text.
Discussion
There is good articulation between results and existing literature, but discrepancies between studies and possible reasons for contextual differences could be explored further.
We thank the reviewer for this constructive suggestion. We have expanded the Discussion section by addressing potential contextual factors that may account for the heterogeneity observed across studies.
There is a lack of discussion on the methodological quality of the included studies (even in a scoping review, it is possible to comment on robustness and limitations).
We thank the reviewer for this valuable observation. We have added a section in the limitation section highlighting the methodological quality of the included studies
Implications for practice could be illustrated with concrete management practices and models already tested.
We thank the reviewer for this valuable suggestion. We have expanded the Implications for Practice section by illustrating concrete and already tested management models
Conclusion
The conclusion summarises the findings well, but could indicate immediate priorities for implementation and key topics for future research. Suggestions such as “adopting participatory models” or “reorganising workload” are broad; it would be more useful to indicate specific tools, frameworks, or policies.
We thank the reviewer for this valuable suggestion. The conclusion has been revised to indicate immediate priorities for implementation and to reference specific organizational models and educational strategies already tested in healthcare. We also clarified key topics for future research.

Reviewer 3 Report
Comments and Suggestions for Authors
Thanks for your work. I have some comments below:
-
I do not have access to the search strategy and tables—please provide them.
-
Please include an overview of oncology nurses’ working environments and explain why this is an important area of research.
-
There appears to be no additional check based on the reference list of included reports—please clarify.
-
As far as I know, there are already many reviews on oncology nurse work and practice-related research, so including only 20 reports may be inappropriate. I suggest consulting a librarian to check your search keywords.
-
The discussion sometimes reads like a list of observations from the literature. Consider organizing it under clear thematic subheadings (e.g., Leadership and Governance, Work Environment and Patient Safety, Psychosocial Well-being and Retention) to better synthesize findings.
-
Several points are descriptive (e.g., “leadership is central,” “supportive environments improve retention”) but would benefit from mechanistic explanation—why and how these factors influence outcomes. For instance, explain how transformational leadership mitigates burnout (e.g., via empowerment, trust-building, workload management).
-
The link between missed nursing care, ethical climate, and patient safety is briefly mentioned—please elaborate with conceptual or theoretical underpinnings (e.g., Donabedian’s framework, moral distress literature).
-
While the heterogeneity of contexts is acknowledged as a limitation, the discussion could be strengthened by suggesting how findings might be adapted to different healthcare systems (e.g., low-resource vs. high-resource oncology settings).
-
The recommendations for integrating resilience, communication, and leadership training into curricula are strong—make them more persuasive by referencing competency frameworks (e.g., ICN, AACN Essentials) or successful models from other specialties.
-
Some sentences are overly long and could be streamlined for readability.
-
Example: “The emotional and organizational intensity of oncology care, especially in end-of-life settings, underscores the need for environments that are not only well-resourced and structurally sound, but also psychologically supportive” → could be tightened without losing meaning.
-
Author Response
Rev 3
Thanks for your work. I have some comments below:
- I do not have access to the search strategy and tables—please provide them.
We thank the reviewer for the observation. The complete search strategy and the extraction tables of the 20 included studies have been attached as Supplementary Materials and are available for consultation.
- Please include an overview of oncology nurses’ working environments and explain why this is an important area of research.
We thank the reviewer for this valuable suggestion. The Introduction has been revised to include a clearer overview of oncology nurses’ working environments and to better emphasize why this represents a critical area of research.
- There appears to be no additional check based on the reference list of included reports—please clarify.
We thank the reviewer for this valuable observation. We confirm that the reference lists of the included reports were also screened to identify additional relevant sources, and this aspect has now been explicitly added to the Methods section.
- As far as I know, there are already many reviews on oncology nurse work and practice-related research, so including only 20 reports may be inappropriate. I suggest consulting a librarian to check your search keywords.
Thank you for your comment. We confirm that the search strategy was developed and reviewed in consultation with a biomedical librarian to ensure its robustness. For clarity, we have now better clarified the inclusion/exclusion criteria in the method section.
- The discussion sometimes reads like a list of observations from the literature. Consider organizing it under clear thematic subheadings (e.g., Leadership and Governance, Work Environment and Patient Safety, Psychosocial Well-being and Retention) to better synthesize findings.
Thank you for this helpful comment. The discussion has been revised to improve synthesis and coherence. However, we did not introduce thematic subheadings, as this would have overlapped with Results section. Instead, we aimed to deepen the interpretation of selected aspects without replicating the mapping of evidence already provided.
- Several points are descriptive (e.g., “leadership is central,” “supportive environments improve retention”) but would benefit from mechanistic explanation—why and how these factors influence outcomes. For instance, explain how transformational leadership mitigates burnout (e.g., via empowerment, trust-building, workload management).
Thank you very much for this insightful suggestion. We have integrated this point into the Discussion.
- The link between missed nursing care, ethical climate, and patient safety is briefly mentioned—please elaborate with conceptual or theoretical underpinnings (e.g., Donabedian’s framework, moral distress literature).
Thank you very much for this valuable suggestion. We have addressed it by expanding the discussion and explicitly elaborating on the link between missed nursing care, ethical climate, and patient safety. In particular, we integrated conceptual underpinnings from Donabedian’s framework.
- While the heterogeneity of contexts is acknowledged as a limitation, the discussion could be strengthened by suggesting how findings might be adapted to different healthcare systems (e.g., low-resource vs. high-resource oncology settings).
We thank the reviewer for this important observation. The Discussion has been revised to address how the findings could be adapted to different healthcare systems
- The recommendations for integrating resilience, communication, and leadership training into curricula are strong—make them more persuasive by referencing competency frameworks (e.g., ICN, AACN Essentials) or successful models from other specialties.
Thank you for this valuable suggestion. As requested, we have integrated references to the International Council of Nurses (ICN) Framework of Competencies and the American Association of Colleges of Nursing (AACN) Essentials to strengthen the persuasiveness of our results.
- Some sentences are overly long and could be streamlined for readability.
- Example: “The emotional and organizational intensity of oncology care, especially in end-of-life settings, underscores the need for environments that are not only well-resourced and structurally sound, but also psychologically supportive”→ could be tightened without losing meaning.
Thank you for this observation. We revised the highlighted sentence, making it clearer and more concise. In addition, the entire manuscript has been reviewed with particular attention to sentence length, in order to improve overall readability and flow.

Reviewer 4 Report
Comments and Suggestions for Authors
Dear Authors
Please, see few comments you might consider or clarify:
Primary and secondary evidence are pooled without overlap management. You include integrative/narrative/systematic reviews and a metasynthesis as included studies, alongside primary studies, then summarize “frequencies” across the set. This risks double-counting findings and inflating signals.
Separate primary studies from secondary/tertiary evidence; if reviews are retained, run an overlap analysis (e.g., citation matrix/covered area) and synthesize them in an “umbrella” subsection, not in the same counts.
( u use PRISMSA systematic review, correct to PRISMA scoping review)
Inflated and conceptually invalid “average sample size.” You report “an average of 736 nurse participants per study,” despite including qualitative studies and reviews for which “sample size” is not comparable—and reviews have no participants.
Remove the average or compute it solely across eligible primary quantitative studies, reporting median and IQR; do not commingle designs.
Setting/context mismatch (undercoverage). PCC includes hospitals, homes, and community, but results are essentially hospital-only (one university setting). This narrows the map relative to your stated context.
Either broaden retrieval to capture community/home oncology nursing or revise the PCC and title to match the actual scope.
Recency gap. Searches ran in Jan–Apr 2025, yet included studies stop at 2023 without explanation.
Re-run the search to capture 2024–2025 outputs, or explicitly state that screening found none meeting criteria post-2023 which need strict confirmation_
Outcome claims pinned to the wrong source. You attribute reductions in nosocomial infections to Lake 2002—the scale development paper—rather than outcomes research.Re-attribute outcomes to appropriate studies (e.g., outcomes analyses), and keep Lake (2002) for instrument provenance only.
RN4CAST is presented as oncology evidence. You portray RN4CAST (12 countries, >30k nurses) as supporting oncology-specific claims, then cite a Swedish oncology sub-study. The cross-national RN4CAST corpus is not oncology-specific. Clarify that RN4CAST is general acute-care nursing; if you use it, frame it as contextual evidence and avoid implying oncology specificity.
Primary Nursing “well-established … [2,11,35]”: mis-sourced. Lake (2002) again cannot substantiate model adoption; it’s an instrument paper.
(cf. Lake is ref 2:
) Replace with sources documenting Primary Nursing use in oncology or drop the claim.
Overstated consistency of effects. The abstract asserts “a positive NWE was consistently associated with improved … outcomes,” despite heterogeneity and predominance of cross-sectional designs. Temper to “frequently reported associations” and separate nurse- and patient-level outcomes with caveats about design limits.
Best wishes
Author Response
REV4
Dear Authors
Please, see few comments you might consider or clarify:
Primary and secondary evidence are pooled without overlap management. You include integrative/narrative/systematic reviews and a metasynthesis as included studies, alongside primary studies, then summarize “frequencies” across the set. This risks double-counting findings and inflating signals.
Separate primary studies from secondary/tertiary evidence; if reviews are retained, run an overlap analysis (e.g., citation matrix/covered area) and synthesize them in an “umbrella” subsection, not in the same counts.
( u use PRISMSA systematic review, correct to PRISMA scoping review)
We thank the reviewer for this valuable observation. In our scoping review, we included both primary and secondary sources. Primary studies already included in secondary reviews were considered only when, within those reviews, they were analyzed with a different objective from our research question. This clarification has now been added to the Methods section to make our inclusion criteria and the management of potential overlap more explicit. For the reporting of this review, we followed the PRISMA-ScR checklist, as recommended by the Joanna Briggs Institute Manual for Evidence Synthesis (2024). During the selection process, we used the PRISMA 2020 Flow Diagram, originally developed for systematic reviews and subsequently adapted for other types of reviews.
Inflated and conceptually invalid “average sample size.” You report “an average of 736 nurse participants per study,” despite including qualitative studies and reviews for which “sample size” is not comparable—and reviews have no participants.
Remove the average or compute it solely across eligible primary quantitative studies, reporting median and IQR; do not commingle designs.
Thank you very much for this observation. We have removed the reference to the “average sample size” as suggested.
Setting/context mismatch (undercoverage). PCC includes hospitals, homes, and community, but results are essentially hospital-only (one university setting). This narrows the map relative to your stated context.
Either broaden retrieval to capture community/home oncology nursing or revise the PCC and title to match the actual scope.
We thank the reviewer for this insightful comment. As this is a scoping review, our intention was to map the available evidence across hospital, community, and home oncology settings. However, the retrieval process revealed that almost all of the included studies were conducted in hospital contexts, with a clear underrepresentation of home- and community-based oncology nursing. Rather than revising the PCC, we consider this lack of evidence to be an important literature gap that warrants future research. Following the reviewer’s suggestion, we have explicitly addressed this point in the Discussion section, highlighting the need for studies exploring nurses’ work environments in non-hospital oncology settings.
Recency gap. Searches ran in Jan–Apr 2025, yet included studies stop at 2023 without explanation.
Re-run the search to capture 2024–2025 outputs, or explicitly state that screening found none meeting criteria post-2023 which need strict confirmation_
We thank the reviewer for this important observation. no studies published after 2023 met the inclusion criteria for this scoping review. We have now explicitly clarified this aspect in the results section to avoid any ambiguity.
Outcome claims pinned to the wrong source. You attribute reductions in nosocomial infections to Lake 2002—the scale development paper—rather than outcomes research.Re-attribute outcomes to appropriate studies (e.g., outcomes analyses), and keep Lake (2002) for instrument provenance only.
We sincerely apologize for the inappropriate use of the citation. We have carefully replaced it throughout the manuscript and replaced outcomes to appropriate studies.
RN4CAST is presented as oncology evidence. You portray RN4CAST (12 countries, >30k nurses) as supporting oncology-specific claims, then cite a Swedish oncology sub-study. The cross-national RN4CAST corpus is not oncology-specific. Clarify that RN4CAST is general acute-care nursing; if you use it, frame it as contextual evidence and avoid implying oncology specificity.
We thank the reviewer for this accurate observation. We agree that the RN4CAST project as a whole is not oncology-specific, but rather a large cross-national study on acute-care nursing. We have therefore reformulated the text to clarify that the evidence cited in our manuscript refers specifically to the Swedish oncology sub-study derived from the broader RN4CAST project
Primary Nursing “well-established … [2,11,35]”: mis-sourced. Lake (2002) again cannot substantiate model adoption; it’s an instrument paper.
(cf. Lake is ref 2:
) Replace with sources documenting Primary Nursing use in oncology or drop the claim.
Thank you very much for pointing this out. We have carefully revised the text and removed the mis-sourced citation regarding Primary Nursing, ensuring that the statement is now supported only by appropriate references.
Overstated consistency of effects. The abstract asserts “a positive NWE was consistently associated with improved … outcomes,” despite heterogeneity and predominance of cross-sectional designs. Temper to “frequently reported associations” and separate nurse- and patient-level outcomes with caveats about design limits.
Thank you so much. In the abstract, the consistency has been moderated, nurse- and patient-level outcomes were separated, and design limitations explicitly acknowledged, as suggested

Round 2
Reviewer 2 Report
Comments and Suggestions for Authors
After reading the corrections made by the authors, I consider the article suitable for publication.
Reviewer 3 Report
Comments and Suggestions for Authors
Thanks for the revision. I have no further comments.
Reviewer 4 Report
Comments and Suggestions for Authors
Thank you for addressing the comments